# Large-scale cryptic proteome mining revealed potential phage-mediated host-pathogen genetic exchange in *Mycobacterium tuberculosis*

**Nikhil Bhalla**[1,2,3]*, **Nupur Angrish**[3,4]

**1** Department of Medical Epidemiology and Biostatistics, Karolinska Institutet, Stockholm, Sweden,
**2** Translational Health group, International Centre for Genetic Engineering and Biotechnology (ICGEB), New Delhi, India, **3** Department of Biochemistry, University of Delhi South Campus (UDSC), New Delhi, India, **4** Birkbeck, University of London, London, United Kingdom

* nikhilbhalla94@gmail.com

## Abstract

### Background

Due to inevitable evolution, clinical strains of *Mycobacterium tuberculosis* (Mtb) exhibit distinct phenotypes and differ significantly from laboratory strains. This divergence is driven by the acquisition of diverse mutations, intragenomic recombination, and potentially phage-mediated genetic exchange. Further investigation is therefore required to better understand these differences, especially regarding the emergence of novel open reading frames (ORFs).

### Methodology

A large-scale whole-genome sequencing (WGS) dataset from tuberculosis (TB)-endemic countries was assembled into contigs. These contigs were then used to mine the cryptic proteome through *in silico* predictions, using an emerging state-of-the-art deep learning protein language model, ProtBERT. Structures of emerging ORF proteins were predicted using colabfold2. In addition, Ramachandran plot analysis and molecular dynamics simulation (MDS) were performed to assess structural validity and stability.

### Results

Most small cryptic proteins were derived from PE, PPE, and PE-PGRS family genes. Notably, a protein cluster consisting sequences of 101–300 amino acids showed no similarity with the proteins from the reference Mtb H37Rv strain but contained phage-derived domains and sequences homologous to host (primates) DNA. Sub-stratification of this cluster revealed the presence of domains like reverse transcriptases and other phage-associated proteins. BLASTp hits of these proteins showed similarity between these proteins and the host proteome. Structural

**Data availability statement:** All data generated is contained within the paper and associated Supporting Information files. Cluster-19 protein sequences, UMAP coordinates and cluster annotation can be accessed from the Zenodo repository: https://zenodo.org/records/19552619.

**Funding:** The author(s) received no specific funding for this work.

**Competing interests:** The authors declare no conflict of interest.

validation showed that the Phi/Psi angles of modelled proteins were within the accepted ranges. Moreover, MDS of medoids of subclusters displayed stable root mean square deviation (RMSD) and radius of gyration (RGYR) profiles, supporting the structural plausibility of these proteins. Importantly, one sub-cluster showed a higher presence in drug-resistant Mtb strains. The co-occurrence of phage-related domains and host DNA strongly suggests illegitimate phage-mediated lateral transfer of host nucleic acids into the genome of Mtb.

## Conclusion

The majority of the small ORFs are found to be nested within annotated genes. Particular emphasis has been given to the incidental finding of phage-host chimeric ORF signatures within the Mtb genome. This study provides computational evidence supporting the structural stability of these proteins. Thus, it can be speculated that such proteins may contribute to pathogenicity, survival within the host, or molecular mimicry mechanisms.

## Introduction

According to the World Health Organization (WHO) Global Tuberculosis (TB) report, there has been a continuous rise in TB incidence, largely attributed to COVID-19-associated disruptions, which have hindered progress toward global TB eradication goals. TB remains endemic in low- to middle-income tropical nations where health infrastructure is often strained [1]. Several risk factors for TB burden include environmental, genetic, and psychosocial factors such as poverty, crowded living conditions, diabetes, HIV infection, epigenetic modifications, and even compromized hygiene and sanitation [2–5]. Standard TB treatment generally lasts for 6 months; however, the emergence of drug-resistant TB significantly prolongs the treatment duration and is often associated with poor prognosis. This extended treatment leads to compliance fatigue and irregular adherence, thereby increasing the risk of developing more severe and potentially fatal forms of TB with outbreak potential [6–9].

*Mycobacterium tuberculosis* (Mtb), the pathogen that causes TB, is a slow-growing bacterium that has evolved into multiple drug-resistant forms and diversified into various lineages, each with distinct phenotypes, genotypes, and clinical manifestations. This diversity in clinical isolates has been widely reported [10–14]. However, laboratory strains of Mtb, such as H37Rv exhibit comparatively reduced evolutionary diversity due to controlled laboratory conditions and repeated passaging from the same stock. While this standardizes the research procedures, making its use more predictable, clinical isolates continue to evolve dynamically. Consequently, the differences between clinical isolates and laboratory strains extend beyond minor genetic or phenotypic variations [15,16]. These differences manifest in the form of Mtb lineages, differences in the ability to develop drug resistance, lineage-specific pathophysiology, virulence, and spread [11].

Like other bacteria, evolution in Mtb is caused by several mechanisms, such as single-nucleotide polymorphisms (SNPs), insertions/deletions (INDELs), intragenomic

recombination, and mycobacteriophage attacks [17–19]. However, unlike many other bacteria, intra-species horizontal gene transfer (HGT) is rarely reported and remains a topic of debate due to the organism's complex, lipid rich and relatively impermeable cell wall structure [17, 20]. In contrast, gene families such as insertion sequences, CRISPR-associated elements, and toxin-antitoxin loci, many of which have viral origin, indicate ongoing acquisition into the Mtb genome, highlighting that its evolution is driven not only by mutations, albeit less frequently, by phage-mediated gene transfer events [21–23]. This sequence transfer from phages to Mtb is leveraged to produce genetically modified Mtb strains and has been explored as a novel diagnostic approach [24–28].

Events like phage-driven HGT and intra-genome recombination can lead to the introduction of non-Mtb genetic material into the bacterial genome when phage lysogenizes the host and integrates foreign DNA. These events can result in the formation of fusion chimeric genes within the Mtb genome [25,29]. Phages often serve as genetic shuttles, carrying DNA from various hosts, including those found in the microbiota of the Mtb-susceptible hosts, thereby introducing heterologous sequences into Mtb [30]. These chimeric introductions may give rise to uncharacterized or hypothetical genes that encode functional proteins [31]. The emergence and translation of such cryptic ORFs in clinical Mtb isolates is a plausible evolutionary outcome. These novel proteins may play roles in immune evasion through molecular mimicry, virulence, or even the development of drug resistance. In brief, molecular mimicry is when a pathogen, in our case Mtb, encodes host-like proteins and evades the detrimental effects of the host immunity, a strategy well reported in other bacteria and viruses [32–35]. Apart from phage attacks, convergent evolution may also result in the development of host-like proteins, further contributing to molecular mimicry [32,35]. These mimicry-related proteins may be encoded by small ORFs, and isolated literature already points to the presence of small ORFs (encoding <100 amino acid long proteins) in clinical Mtb isolates [36,37]. However, in addition to these smaller elements, longer ORFs may also arise because of continuous evolutionary pressures. As Mtb continues to evolve in the clinical environment, investigating these emerging ORFs is imperative to better understand strain-specific adaptations.

Advances in technologies such as deep learning-based protein embedding models, the availability of WGS datasets of clinical isolates, and cutting-edge bioinformatics tools allow large-scale genome mining of clinical isolates. In the present study, we explore an exploratory computational approach to identify emerging ORFs in the clinical Mtb genome. This involved analysis of a large-scale WGS dataset, cryptic proteome prediction and characterization using embeddings, and structural validation using Colabfold2, molecular dynamics simulations, phi/psi angle determination, and sequence remapping. A key incidental finding of this study was the presence of singleton contigs harboring phage domains and host DNA-like sequences. These features suggest potential phage-mediated lateral gene transfer events contributing to previously unrecognized molecular mimicry mechanisms in Mtb. These findings underscore the need for deeper investigation into the cryptic and chimeric proteome of Mtb clinical isolates.

## Materials and methods

### Data acquisition and pre-processing

WGS datasets previously shortlisted in an earlier study [38] were acquired in fastq format. Quality control, including filtering and trimming, was performed using bbduk.sh by bbtools (https://github.com/BioInfoTools/BBMap) [39]. The resulting high-quality datasets were then subjected to *de novo* assembly using Megahit (https://github.com/voutcn/megahit) (v1.2.9) [40]. The quality of assemblies was assessed using Quast (https://github.com/ablab/quast) (v5.3.0) [41].

### Mining approach

The assemblies were functionally annotated using bakta (https://github.com/oschwengers/bakta) (v1.11.3) with the light database [42]. The hypothetical proteins from all assemblies were pooled and deduplicated using seqkit rmdup (https://github.com/shenwei356/seqkit) (v2.3.0) [43]. To further reduce the redundancy, the deduplicated hypothetical protein

sequences were clustered using cd-hit (https://github.com/weizhongli/cdhit) (v4.8.1) with a sequence identity threshold (-c) of 0.9 and a word length (-n) of 5 [44]. The resulting cluster sequences produced by cd-hit were split into bins based on the protein length. Each bin was embedded independently using the Bio-Embeddings tool (https://github.com/sacdallago/bio_embeddings) with the prottrans_bert_bfd model, following the associated computational protocol and a batch size of four [45,46]. UMAP projections were determined using the prottrans_embeddings protocol with parameters: neighbour = 15, min_dist = 0.1, and the Euclidean metric. The UMAP projections were then subjected to K-means clustering, for which the value of K was determined using the elbow method. For this, custom scripts were used (Github:nikhilbhalla94/OmicHack_Utils/blob/main/kvalue.py;Github:nikhilbhalla94/OmicHack_Utils/blob/main/clustering.py). Twenty proteins were selected randomly from each cluster and subjected to BLASTp against Mtb (Reference: NC_000962.3) protein sequences. The resulting hits were used to annotate the clusters. For the clusters that showed a mix of hits upon BLASTp, the associated gene names were analyzed using ShinyGO for gene ontology (GO) enrichment, and the most enriched GO term was used for annotation [47].

### Cluster sub-stratification, functional analysis, protein structure, and stability prediction

Proteins within the cluster of interest were extracted and subjected to mining and k-mer clustering using the same methodology described in the 'Mining approach' section. Three medoid proteins from the clusters of interest were analyzed using the Interproscan online server for the identification of functional domains [48]. One representative medoid from each cluster of interest was further subjected to three-dimensional (3D) structural modelling using an AlphaFold2-dependent approach with Colabfold-batch (https://github.com/sokrypton/ColabFold), using the –multimer parameter flag [49,50]. Rank one Colabfold2 structures were subsequently processed with colabfold_relax, which added hydrogen atoms and further relaxed the protein structure. The Ramachandraw python library (https://pypi.org/project/ramachandraw/) was used for plotting the Ramachandran plot to assess whether the amino acid psi/phi bonds fell within the acceptable regions [51]. Further, MD simulations were performed using Gromacs (v2025.2) [52]. For this, Amber99sb-ildn force field and tip3p model were employed following a standard "protein in water" procedure, with each simulation run for 20 nanoseconds (link to script: https://github.com/nikhilbhalla94/OmicHack_Utils/blob/main/gromacs.sh).

### Drug resistance and lineage profiling

To determine the abundance of drug resistance among sensitive clinical isolates within clusters of particular interest, TB-Profiler (https://github.com/jodyphelan/TBProfiler) (v6.6.5) was used to determine drug resistance and lineage profiles. Relative abundance was visualized using stacked bar plots [53].

### Validation by remapping of raw reads

Medoid protein corresponding to the WGS dataset in fastq format was mapped to the nucleotide sequences of medoid proteins using the BWA-MEM algorithm (v0.7.17.r1188) [54]. The mapped files were sorted by coordinate, and the read depth was determined using samtools (v1.19.2) [55]. The contamination and genome completion metrics of genomes of interest was carried out with checkM2 v1.1.1.

### Computational resources

All analysis was conducted on a consumer-grade customized computer running WSL-2 on an AMD 5600G chipset, 64 GB RAM, and NVIDIA GeForce GTX 1650 GPU. For embedding, ColabFold2, and MD simulations, unique versions of GPU and CUDA drivers were used depending on the compatibility of the tools. All genomic analyses (viz., Adapter trimming, de novo assembly, QC, and functional annotation) were performed on the CPU, and studies such as embedding, detecting UMAP projections, 3D structure prediction, and MD simulations were carried out by leveraging NVIDIA GPU-accelerated

parallel computing. Since the GPU used in the present study has limited VRAM (4 GB), batch size was reduced from 4 to 1 whenever necessary, automatically by Bioembeddings, and this change only impacted the turnaround time. Wherever feasible, parallel computing was leveraged using GNU parallel [56].

## Results

### Assembly quality and QC metrics

Sample sizes, before and after processing: In total, 8224 WGS datasets, primarily from TB-endemic regions of the world, were analyzed in the present study. The SRA accession numbers, along with de novo assembly metrics, are presented in S1 Table.

### Binning and mining of in silico predicted cryptic proteome

After pooling hypothetical proteins identified through functional annotation, the total protein count, which included duplicate and redundant (based on similarity) proteins, was ~6.8 million (n = 6775995). Deduplication reduced this number to ~1 million (n = 975806), and subsequent redundancy using cd-hit further decreased it to 0.44 million (n = 445596). Binning of proteins was carried out based on their lengths, resulting in the following bins with unique protein numbers: 1–100, 101–300 amino acids (AA) (n = 217140), 301–500 AA (n = 218584), 501–700 AA (n = 1690), 701–900 AA (n = 701), 901–1100 AA (n = 342) and 1101–1300 (n = 200) and 1301–4100 AA (n = 454). Inertia vs K-value plots were used manually to determine the elbow point. Slightly higher than the elbow point K value, where the inertia decrease is mitigated considerably with increasing K-values. This was done to enhance the clustering sensitivity and allow for finer sub-stratification. The inertia vs K plots are presented in S1 Fig.

**Bin-1** proteins, containing small ORFs of up to 100 AA, were mined using an embedding approach with the prottrans_bert_bfdmodel. Subsequent clustering of UMAP projections by K-means (k = 20) resulted in 20 distinct clusters. BLASTp analysis of 20 randomly selected proteins from these clusters identified major hits associated with PE-PGRS and PPE regions, indicating that these proteins are nested ORFs within the well-characterized genes. Other clusters also showed partial similarity to genes involved in lipid biosynthesis, the polyketide synthase (pks) system, ClpB, PurK, integral membrane proteins (IMPs), and known Mtb transposases. In addition, 3 clusters of unknown origin were identified: Unknown_5, Unknown_14, and Unknown_18, having 5131, 753, and 10570 deduplicated protein sequences, respectively, derived from multiple samples (Fig 1A).

**Bin-2**, comprising proteins ranging from 101–300 AA after embedding and k-means clustering (k = 20), resulted in 18 clusters with identifiable biological relevance. These clusters correspond to known nested regions or functional categories, including amino acid catabolism, PE-PGRS/wag, HNH endonuclease, PPE family proteins, Pks/VapC/MetK, Cut/Mce/Ggt/Zmp, PE-PGRS/Pks/Mms/Rpf, PPE/CdaR/PucR, transposase, DUF2710/GabD/PE-PGRS/Wag22, FadD11/Rv3633, Rv2417c, PE-PGRS/Wag, PPE/Esx, and complex multifunctional clusters such as Pks/FadD/ArgH/CtpF/GlgP/RpmG1/Msh/ Tcr. Additionally, two clusters were associated with integral membrane proteins (IMPs), one mapping to DctA (Rv0955) and another to QcrA (Rv0986).

Moreover, 3 clusters designated as Unknown_5, Unknown_7, and Unknown_19 with 12186, 1594, and 971 nonredundant proteins, respectively, did not match with any known Mtb protein in BLASTp searches (Fig 1B).

**Bin-3,** consisting of 301–500 AA proteins after embedding and k-means clustering (k = 15), resulted in 14 clusters. These clusters correspond to a diverse set of characterized Mtb proteins, such as Transposases, Fad proteins, MalQ CobD, AcrA, Aks, Aas; PE-PGRS/Wag; PPE/PE-PGRS/Wag; Porins, and membrane-associated proteins, such as PE-PGRS, Rv2797c, Rv3897c-98c, Rv3903c, EccB, EspA-J-K, LipY, PknH. Other clusters mapped to proteins involved in sphingolipid metabolism or represent mixed categories, including Ecc, Eis, Ggt, Hyc, Lip, and Ltp proteins. Additional matches included hypothetical or poorly characterized proteins such as Rv0029, Rv1576c, Rv2650c, Rv2405, Rv3363c;

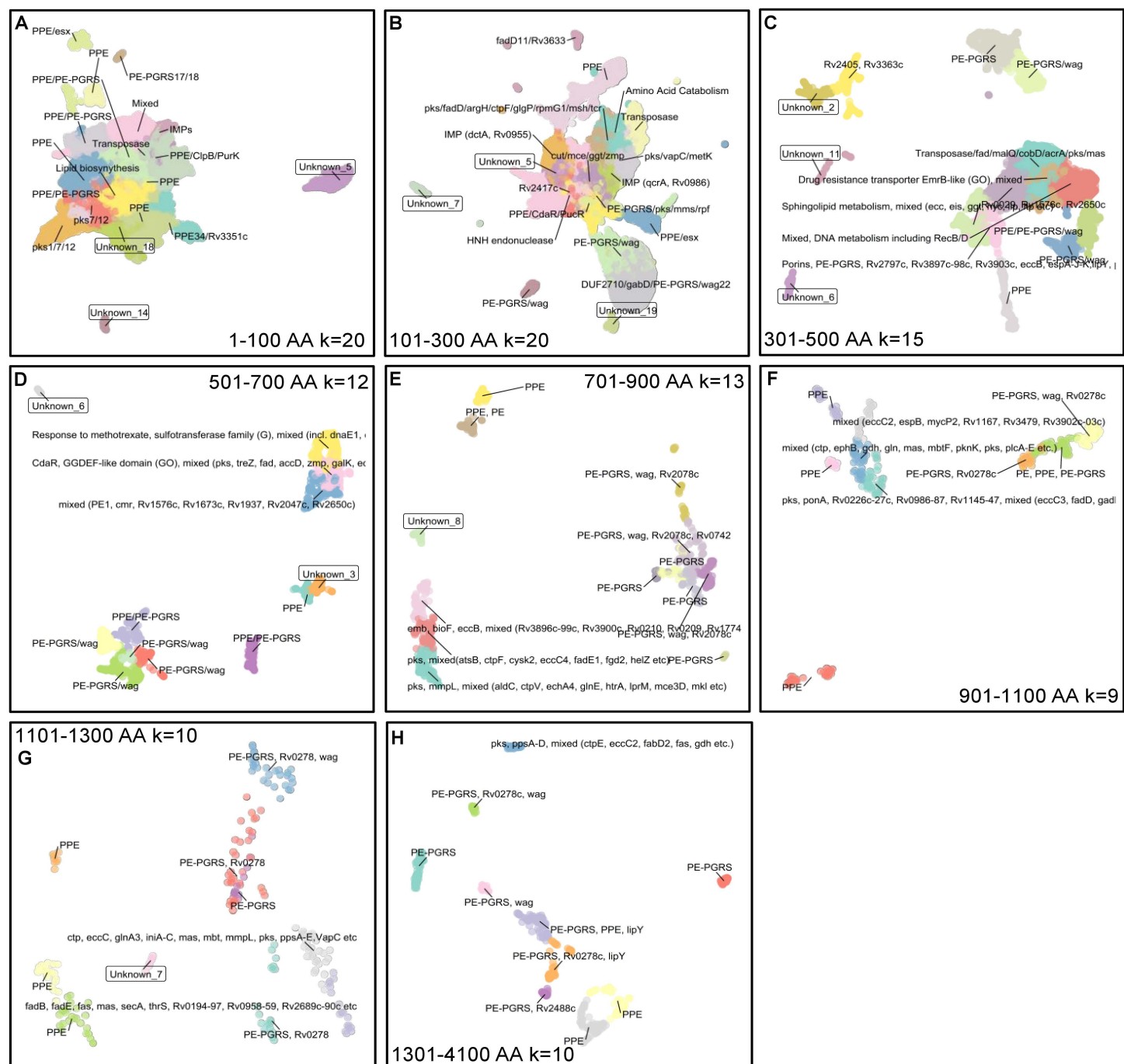

**Fig 1. Embedding hypotheticals revealed a signature of highly divergent homologs or non-Mtb-derived proteins.** Publicly available WGS datasets (n = 8224) were *de novo* assembled and functionally annotated. The hypothetical predictions made during the functional annotation step were deduplicated using cd-hit. The hypothetical proteins were split into the bins of 1-100, 101-300, 301-500, 501-700, 701-900, 901-1100, 1101-1300, and 1301-4100 amino acids. The proteins in each bin were subjected to embedding using the model prottrans_bert_bfd. The resulting UMAP projections were clustered using K-means, and the value of K was determined using the Elbow method. To increase the sensitivity of clustering and substratification, K, as mentioned on the UMAP, was used for clustering. From each cluster, twenty randomly selected proteins were extracted and subjected to pairwise alignment with the proteins of Mtb (Ref: NC_000962.3), and the hits were used for annotating the clusters. Gene ontology cluster terms were used to annotate the clusters that showed similarity to multiple Mtb proteins. The clusters that showed no pairwise alignment were annotated as Unknowns. For calculating UMAP projections, prottrans_embeddings with neighbors = 15, min_dist = 0.1, and Euclidean metric were used.

clusters associated with DNA metabolism (e.g., RecB/D) and drug resistance-related transporters such as EmrB-like proteins. A number of clusters also showed strong association with PE-PGRS/Wag protein family and other mixed functional categories. Besides, three clusters which showed no similarity with Mtb proteins are designated as Unknown_2, Unknown_6, and Unknown_11, with 331, 119, and 89 proteins, respectively, as shown in Fig 1C.

**Bin-4** with proteins of 501–700 AA in length showed 11 clusters with a K value of 15. The clusters were associated with a range of Mtb proteins and functional categories, including: PPE; PE-PGRS/Wag; mixed proteins such as PE1, Cmr, Rv1576c, Rv1673c, Rv1937, Rv2047c, Rv2650c. Additional clusters showed overlap with CdaR, GGDEF-like domain (as identified by GO), proteins involved in complex metabolic or regulatory pathways, including Pks, TreZ, Fad, AccD, Zmp, GalK, and EccD. Other clusters were categorized as PPE/PE-PGRS, and one cluster was associated with genes responsive to methotrexate/sulfotransferase family (group G), as well as additional mixed function genes, including dnaE1and ctp. Two unknown clusters (unknown_6/3) with 23 and 115 proteins, respectively (Fig 1, second row and second column), were also identified (Fig 1D).

**Bin-5,** comprising proteins of 701–900 long AA after embedding and K-means clustering (k = 13), resulted in 14 clusters that matched primarily with portions and hopped over to neighboring genes of PE, PPE, and PE-PGRS, wag, Rv2078, Rv0742, emb operon, bioF, eccB, pks system, and mmpL. It also identified one distinct and unknown cluster (unknown_8) with 29 proteins that showed no matches with Mtb proteins, as depicted in Fig 1E.

**Bin-6, 7, and 8** showed 8, 9, and 9 clusters with predominance of PE-PGRS, PPE-derived proteins, and some proteins of the Pks family. Only one unknown cluster of seven proteins was found in bin-7, as shown in Fig 1G. Bin-6 and 8 are shown in Fig 1F and 1H.

## Cluster-19 (bin-2) showed >60% identity with the host genes that contained phage-derived domains

To characterize the unknown clusters, Interproscan domain and family profiling was performed which showed extensive hits with phage-derived domains such as L1 transposable element, retrotransposable element, L1 endonuclease, RBD domain, trimerization domain, reverse transcriptase, and domains of similar profiles. Other unknown clusters contained domains representative of potential contamination (Fig 2A). Cluster 19 of bin 2 was sub-stratified, and it was found that it consisted of 8 sub-clusters, medoids of which showed hits with reverse transcriptase, GAG-POL polyprotein, diguanylate cyclase, CDP, exonuclease, and phosphatase with InterProScan (Fig 2B). In addition, BLASTp analysis using a non-redundant database was performed to investigate the origins and conservation of sub-cluster medoid sequences. Sub-cluster zero exhibited sequence similarity to proteins from *Helicobacter pylori*, *Homo sapiens*, and *Macaca mulatta*, suggesting potential molecular convergence, host mimicry, or phage-driven horizontal gene transfer. Sub-clusters 1 and 4 with predicted intrinsic disorder (MobiDB-lite) showed homology to *Rhizopus delemar* (a fungal species) and some fish proteins, indicating high conservation outside Mtb species but absent in Mtb H37Rv reference genome.

Sub-cluster-2 medoids were characterized as reverse transcriptase/endonuclease domains (LINE-1 ORF2p) and demonstrated sequence similarity with the proteins of primates, common hosts of the Mtb complex (MTBC). Similarly, sub-cluster-3 medoids contained domains of reverse transcriptase (RNA-dependent DNA polymerase), often associated with phages, and showed sequence similarity with human, *fusobacterium*, and *micronospora* proteins, again pointing to phage-driven conservation or gene flow.

Subcluster 5 medoids displayed features of classical L1 retrotransposable element, including ORF1p, RNA recognition motifs (RRM), and trimerization domains, and showed shared sequence similarity with *homo sapiens*, *Pan troglodytes* (Chimpanzee), and *Helicobacter pylori*. These sub-clusters also possess phage-like domains such as reverse transcriptase, GAG-POL, HIV-1 RT, and polymerase superfamily. However, its medoids showed exclusive similarity to *Phaseolus vulgaris* (plant), likely due to phage conservation.

Sub-cluster-7 medoids contained a mixture of domains, including reverse transcriptase, endonuclease, DNA/RNA polymerase, CRISPR-associated (Cas) systems, and displayed sequence similarity with *Helicobacter pylori*, *Pan troglodytes*

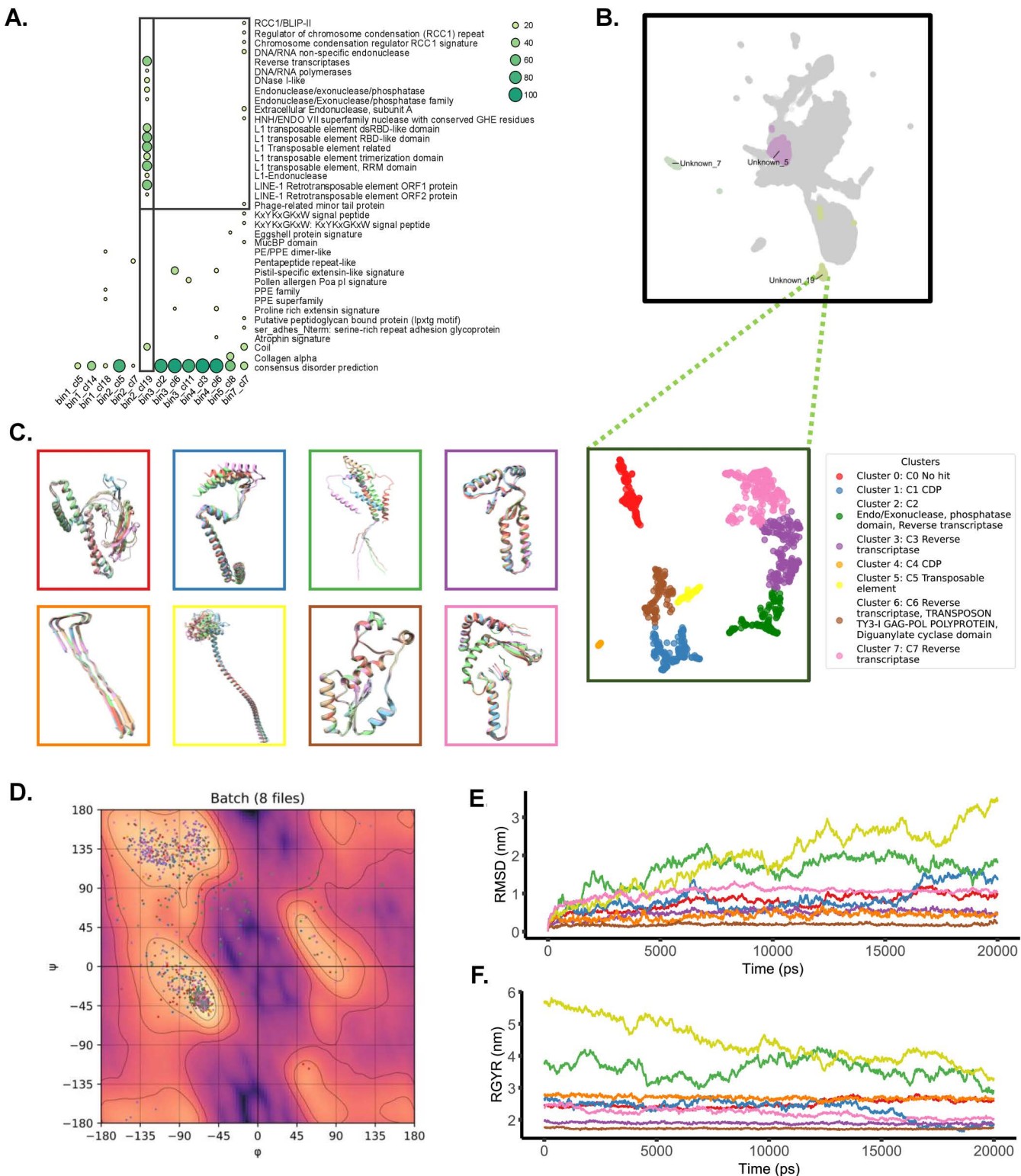

**Fig 2. Emerging ORFs in clinical Mtb isolates were mostly phage-derived and are stable. A:** Representative random sequences from clusters that showed no BLAST hit with Mtb proteins were extracted (n = 20/cluster) and subjected to InterProScan analysis to identify their function and origins. The number of unique proteins per cluster matching the same annotation was counted and normalized to the actual sampling size (n = 20). Percentages were plotted as a dot plot to represent enrichment of specific terms through InterProScan analysis. **B:** Unknown_19 cluster of bin-2 was substratified, and the

medoids of resulting sub-clusters were annotated with InterProScan-determined domains. C: The medoids were also subjected to colabfold2 protein modelling and structures of all ranks (n = 5) were aligned and represented. D: The phi/psi angles of the modelled structures were determined and plotted to determine the plausibility and stability of the structure. E and F: The colabfold2 structures were minimized and subjected to molecular dynamics simulation to determine RMSD and RGYR profiles.

(Chimpanzee), *homo sapiens*, and Cas9-like synthetic constructs. While factors such as convergence and contamination cannot be fully ruled out, the association of phage-like elements and host-derived DNA, especially from primates, within the same medoid is striking and may represent a biologically significant event. A comprehensive summary of InterProScan and BLASTp results is provided in S2 Table.

Colabfold-based structural modeling of medoid protein sequences from identified subclusters revealed distinct 3D structures and unique composition of secondary structure profiles (Fig 2C). The resulting medoid protein 3D conformations exhibited favorable stereochemistry, with most of their Psi/Phi angles falling within the acceptable regions of the Ramachandran plot, supporting the validity of the *in-silico* folding process and the predicted stability of the proteins (Fig 2D). Simple 'protein in water' MD simulations of the medoids of these subclusters displayed stable RMSD profiles except for the medoid of subclusters 2 and 7. Interestingly, the medoid of subcluster 1 showed a stable profile until 15000 ps, and a sudden increase in RMSD was observed (Fig 2E). Analysis of RGYR also showed stable profiles for all proteins except for medoids of subclusters 2 and 7. These two initially displayed higher RGYR values that gradually decreased over the course of the simulation, indicating potential folding rearrangement or delayed stabilization (Fig 2F).

### Sub-cluster 6 of cluster 19, bin 2, shows selective presence in drug-resistant isolates

Three medoids from sub-cluster 19 (bin 2) were further substratified and analyzed according to drug resistance and lineage profiles. Overall, subcluster six was predominantly enriched with isolates from Mtb lineage 1 and 3, while contributions from lineage 2 and 4 were comparatively lower. The majority of subcluster proteins belonged to both drug-resistant and sensitive clinical isolates. Subcluster 1 (indicated in blue) seemed to be contributed maximally by drug-sensitive and fewer drug-resistant lineage 1 and 3 isolates. However, drug-resistant lineage 2 and 4 isolates contributed maximally. From this, it can be speculated that this cluster may have some degree of lineage-specific association with drug resistance status. Specifically, the enrichment of sub-cluster 6 with drug-resistant lineage 2 and 4 isolates highlight its potential role in resistance-related mechanisms (Fig 3).

### Validation of medoids

To validate the authenticity of the unknown cluster containing phage-derived domains and host DNA signatures, the medoid genes of subclusters were mapped against raw sequencing reads. All sub-clusters demonstrated contiguous read coverage, with no regions exhibiting a read depth of zero. However, as expected, some subclusters showed higher read depth than others, owing to their study origin and sequencing coverage targeted by the respective authors. No break in read depth < 1 validates that the genes encoding phage-derived proteins containing host elements are not assembly artefacts and are indeed covered by sequencing reads (Fig 4).

Furthermore, the identified medoid gene sequences were exclusively located on singleton contigs. If the origin of these genes were due to contamination, one would expect them to appear as a part of multi-gene fragments typically associated with contaminant DNA. The consistent presence of these sequences as isolated contigs strengthens the argument of their biological relevance and supports the hypothesis of genomic incorporation through phage-mediated HGT rather than random contamination. Further, we also performed checkM2 analysis for analyzing genome completion and contamination metrics of the cluster 19 genomes (n = 971). This showed 0.3% contamination (median) with values ranging from 0.1 to 4.3% and genome completion of 100% median with values ranging from 94% to 100%. Individual sample details with SRA accession numbers is provided in S3 Table.

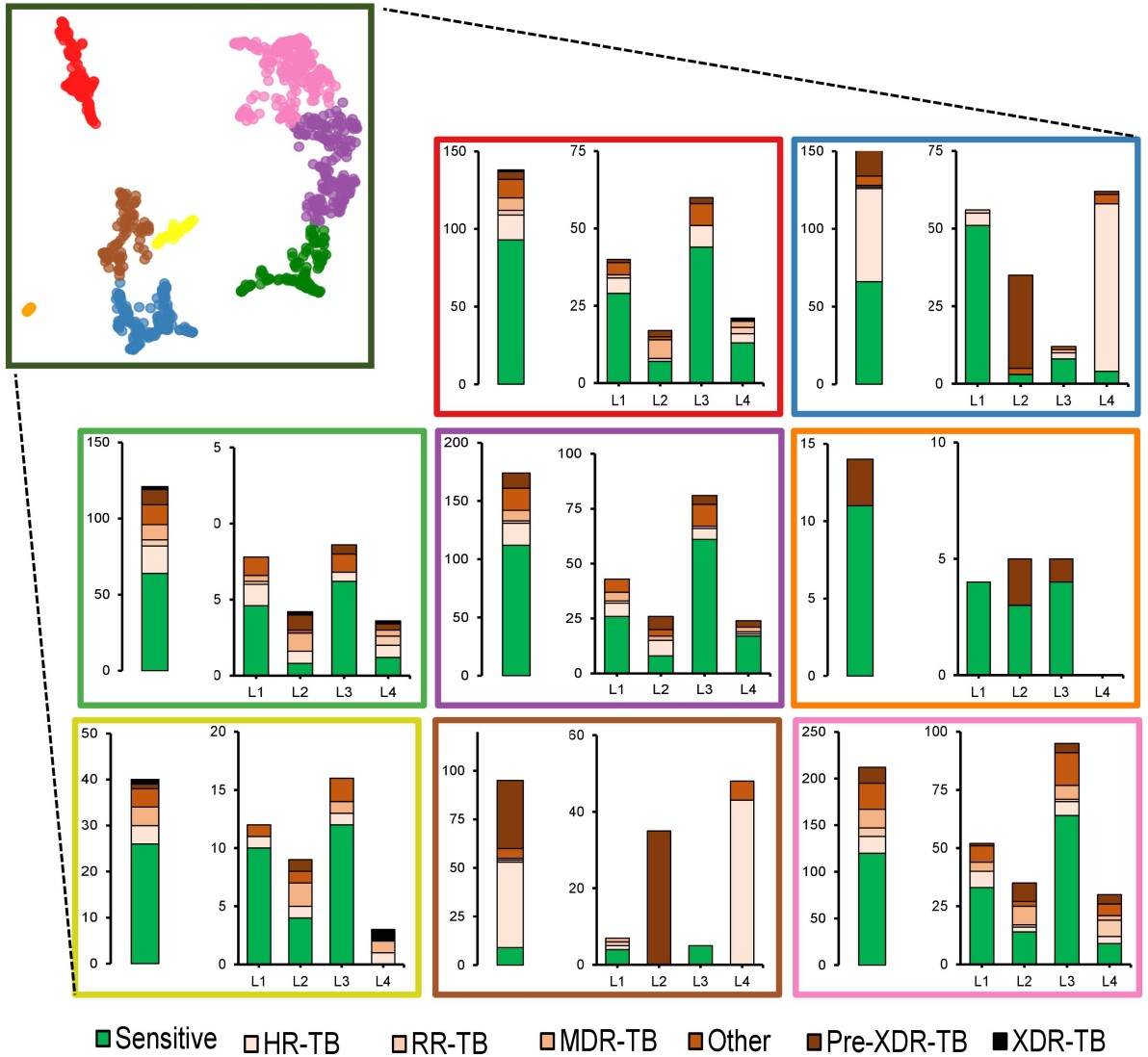

**Fig 3. Remapping and transcriptome-reanalysis assisted validation.** Assemblies that constituted cluster-19 of bin-2 were subjected to analysis with TB-profiler to determine their lineage and specifics of drug resistance status. In each box, an overall distribution is shown separately from the lineage-specific distribution. Abbreviations: HR: Isoniazid resistant, RR: Rifampicin resistant, MDR: Multiple drug resistant, Pre-XDR: pre-Extensively drug resistant, XDR: Extensively drug resistant.

## Discussion

The present study was initially aimed to explore emerging ORFs in clinical Mtb isolates, focusing on genomic features that may be masked in laboratory strains of Mtb due to their reduced evolutionary dynamics. It was anticipated that this exploratory analysis would primarily identify small, previously unannotated ORFs. While such small novel ORFs were indeed identified, sequence similarity analysis revealed that many were nested within well-characterized genes, such as PE, PE-PGRS, and PPE family, indicating that they may represent cryptic coding sequences within known genes.

In other bins, novel ORFs of higher length were also discovered, which showed sequence similarity with well-characterized genes. These findings suggest the possibility of recombination-driven fusions, leading to the formation of

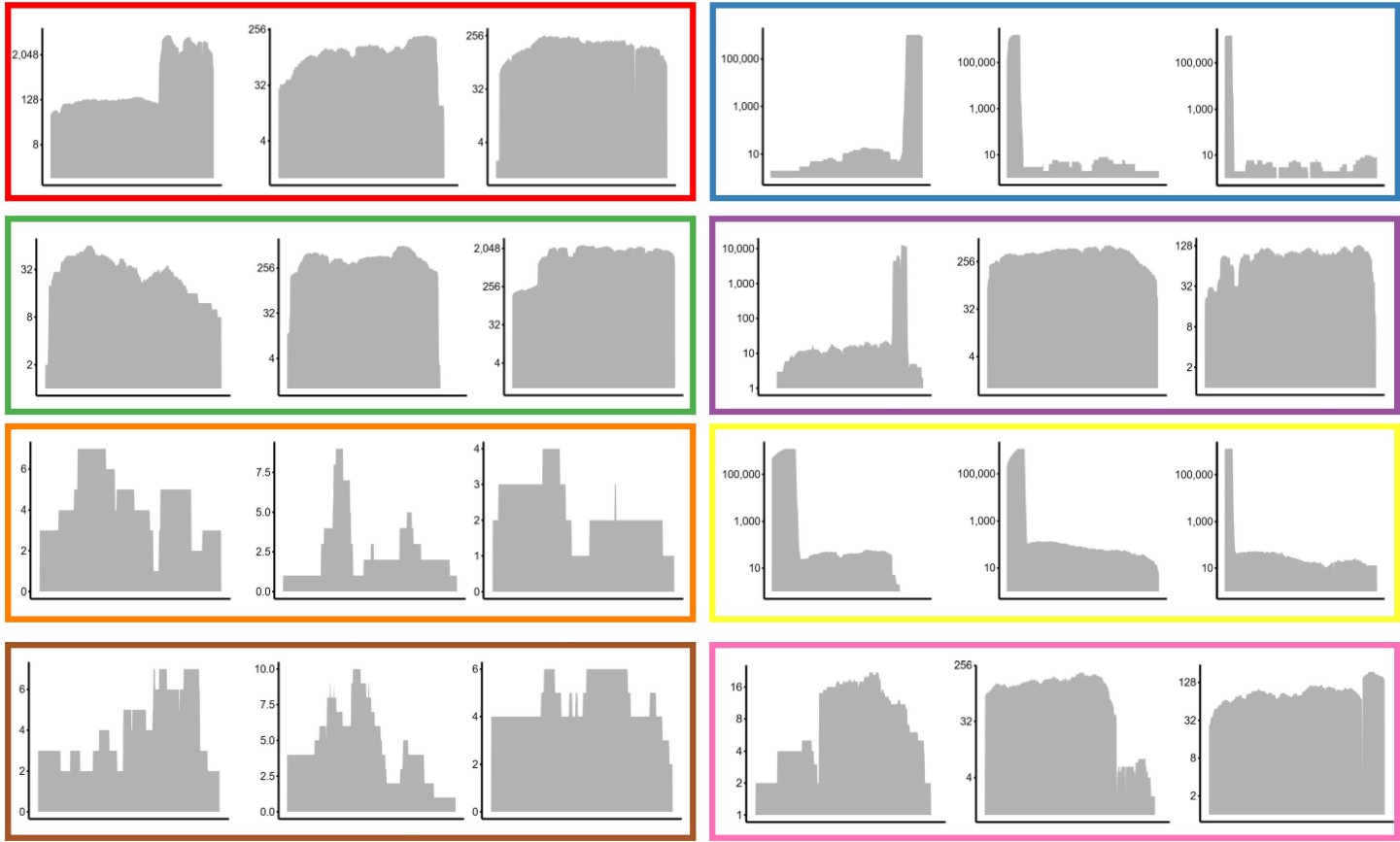

**Fig 4. Validation by remapping of reads to medoid genes.** Reads were mapped using the BWA-MEM algorithm and the medoid gene sequence as a reference. The mapped reads were sorted, and the depth at each locus was determined using the Samtools depth module. For each cluster, three representative medoid genes were taken and mapped with their own WGS data. Details of 3 medoids per subcluster are included in the supplementary. The outline colors represent the subclusters of Cluster-19/bin-2.

potential chimeric genes. Interestingly, some unknown clusters of ORFs showed similarity with distantly related species, suggesting interspecies genetic exchange.

Of particular interest was the identification of a cluster of genes encoding proteins of 101–300 AA in length, which contained domains of phages and host DNA (primates). While horizontal gene transfer is rarely reported in Mtb, the attacks and subsequent lytic lysis of mycobacteriophages are well documented and leveraged in molecular biology techniques to develop knockouts and diagnostics. These phages are also reported to carry host DNA features, and upon lysogenisation, can introduce non-native sequences into the bacterial genome. This possibility is particularly relevant in the context of clinical Mtb isolates, which may be prone to such genetic rearrangements.

A study by Ju et al. has recently reported a high abundance of incomplete transcripts in Mtb H37Rv due to stalled RNA polymerase, which aids in adapting the bacteria to stressful conditions. [57]. Smith et al. reported previously unreported small and nested ORFs using Ribo-seq analysis [58]. Moreover, Barth et al. described the mechanisms of the VapC4 toxin that engages several small ORFs and leads to the regulation of stress responses [36]. However, these scenarios may be attributed to other unknown factors, such as evolution. Phages and intragenomic recombination can introduce non-native genes and lead to the formation of chimeric genes that may have a role in shaping bacterial physiology and host-pathogen interactions.

It is widely known that bacteriophages that primarily infect bacteria can acquire eukaryotic DNA. For instance, Bordenstein et al. discovered the presence of the latrotoxin C-terminal domain, a toxin produced by black widow spiders and other eukaryotic-specific genes/domains in the genome of purified bacteriophage WO viral particles of *Wolbachia*. This study provided strong evidence for phage-mediated acquisition and HGT of eukaryotic DNA [59]. Similarly, other studies have identified eukaryotic genes such as dihydrofolate reductase, thymidylate synthase, and glycine hydroxymethyltransferase in the genome of Hz-1 viruses [60].

Through bioinformatics and molecular analysis, researchers have identified phage-driven horizontal gene transfer in Wolbachia and provided evidence of bacteriophage ability to carry non-native ankyrin domain-containing genes. [61]. Another interesting report on mycobacteriophage Giles, which infects *Mycobacteria*, dairy bacteria, *Staphylococcus*, and *Pseudomonas*, carries mosaic DNA sequences from these hosts. These appear to rise through illegitimate recombination or integrase-mediated site-specific recombination [62]. A study by Chen et al. provided evidence of cross-species phage-mediated HGT of toxin genes between *Staphylococci* and *Listeria*, underscoring phage roles in pathogenicity gene mobility [63].

Phages are recognized as drivers of HGT involving the transfer of genes involved in pathogenicity, virulence, and antimicrobial resistance genes [64]. Even in Mtb H37Rv, several gene cassettes such as Rv1573-Rv1585c and Rv1577c, Rv2645-Rv2659c, annotated as phage-derived in the NC_000962.3 reference genome, exhibit lineage-specific perturbed copy number variations and presence-absence patterns in drug-resistant and sensitive isolates [38,65]. These loci may represent remnants of phage integration events with potential consequences on host-pathogen interactions and bacterial physiology.

In addition to these studies, mycobacteriophages are often detected as components of microbiota and in the exposed tissues of TB-susceptible organisms such as humans and bovines [66,67]. Their presence in these hosts suggests their ecological and evolutionary role in shaping the Mtb genome.

The above-mentioned studies collectively suggest a widespread phenomenon of phage-mediated transfer of genetic material, both intra- and cross-species across various bacterial species, as well as signatures of historical phage-mediated transfers of genetic material in Mtb. Conventionally, mycobacteriophages have been utilized to modify the genomic DNA of Mtb to generate knockouts. Since laboratory strain H37Rv exhibits reduced evolutionary diversity compared to clinical isolates, the findings of the present study, that is, the presence of phage domains linked with host DNA in clinical Mtb genomes, may represent evolutionary events that occurred during natural adaptation. Apart from this, the occurrence of singleton contigs encoding single medoid genes rules out the possibility of any contamination (in the case of contamination, multigene fragments/contigs are expected). Moreover, phage DNA is known to have erratic GC composition and frequently contains transposons, which can interfere with the tagmentation step of WGS library preparation methods by introducing nicks in the genomic DNA. This may partly explain the occurrence of singleton contigs encoding phage-host-like DNA.

Another possibility for the presence of host-like DNA is convergent evolution for developing molecular mimicry mechanisms to evade the host immune system. These findings raise the questions about the mechanisms and evolutionary significance of phage-host DNA interactions in clinical Mtb isolates. Scientists and groups with access to various clinical Mtb isolates are encouraged to investigate these signatures further, with an aim of validating the occurrence and impact of phage-mediated host DNA transfer in Mtb.

## Supporting information

**S1 Fig. UMAP 3D projections in the form of coordinates were subjected to K-means clustering with increasing K values, and inertia (a measure of cluster compactness) was recorded for each K value.**
(DOCX)

**S1 Table. SRA accessions of the WGS datasets used in present study with Lineage and drug resistance profiles and de novo quality metrics.**
(XLSX)

**S2 Table. Domain characterization and sequence similarity of hypothetical, now partially characterized gene clusters.** Representative medoids of sub-clusters after sub-stratification of cluster 19 of bin 2 were subjected to analysis with InterProScan and non-redundant BLASTp, and top hits for each query have been summarized. Note: These medoids showed no hit with BLASTp with M. tuberculosis selected in the species drop-down box to include in the BLASTp search. A: % identity, B: alignment length, C: Mismatches, D: gap opens, E: q. start, F: q. end, G: s. start, H: s. end, I: evalue, J: bit score.
(XLSX)

**S3 Table. CheckM2 analysis of cluster 19 genomes showing genome completeness and contamination details.**
(XLSX)

## Acknowledgments

The open-source tool developers and submitters of Mtb WGS in public repositories are acknowledged.

## Author contributions

**Conceptualization:** Nikhil Bhalla, Nupur Angrish.

**Data curation:** Nikhil Bhalla, Nupur Angrish.

**Formal analysis:** Nikhil Bhalla, Nupur Angrish.

**Investigation:** Nikhil Bhalla, Nupur Angrish.

**Methodology:** Nikhil Bhalla, Nupur Angrish.

**Resources:** Nikhil Bhalla.

**Software:** Nikhil Bhalla.

**Validation:** Nikhil Bhalla, Nupur Angrish.

**Visualization:** Nikhil Bhalla, Nupur Angrish.

**Writing – original draft:** Nikhil Bhalla, Nupur Angrish.

**Writing – review & editing:** Nupur Angrish.

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
