## [Decision Letter · Decision Letter 0]

21 Jan 2026

PONE-D-25-49411Host-like sequences in emerging cryptic phage proteins of clinical Mycobacterium tuberculosis isolates: phage-driven horizontal transfer or convergent evolution?PLOS One

Dear Dr. Bhalla,

Thank you for submitting your manuscript to PLOS ONE. After careful consideration, we feel that it has merit but does not fully meet PLOS ONE’s publication criteria as it currently stands. Therefore, we invite you to submit a revised version of the manuscript that addresses the points raised during the review process.

We look forward to receiving your revised manuscript.

Kind regards,

Swati Jaiswal

Academic Editor

PLOS One

**Journal Requirements:**

1. When submitting your revision, we need you to address these additional requirements. Please ensure that your manuscript meets PLOS ONE's style requirements, including those for file naming. The PLOS ONE style templates can be found at https://journals.plos.org/plosone/s/file?id=wjVg/PLOSOne_formatting_sample_main_body.pdf and https://journals.plos.org/plosone/s/file?id=ba62/PLOSOne_formatting_sample_title_authors_affiliations.pdf 2. Please upload a new copy of Figures 1 to 4 and S1, as the detail is not clear. Please follow the link for more information:  https://journals.plos.org/plosone/s/figures 3. Please include a new copy of Table S2, in your manuscript; the current table is difficult to read. Please follow the link for more information: https://journals.plos.org/plosone/s/tables 4. Please include captions for your Supporting Information files at the end of your manuscript, and update any in-text citations to match accordingly. Please see our Supporting Information guidelines for more information: http://journals.plos.org/plosone/s/supporting-information. 5. If the reviewer comments include a recommendation to cite specific previously published works, please review and evaluate these publications to determine whether they are relevant and should be cited. There is no requirement to cite these works unless the editor has indicated otherwise.

Reviewers' comments:

Reviewer's Responses to Questions

**Comments to the Author**

1. Is the manuscript technically sound, and do the data support the conclusions?

Reviewer #1: Partly

Reviewer #2: Partly

2. Has the statistical analysis been performed appropriately and rigorously? 

Reviewer #1: Yes

Reviewer #2: I Don't Know

3. Have the authors made all data underlying the findings in their manuscript fully available?

Reviewer #1: No

Reviewer #2: Yes

4. Is the manuscript presented in an intelligible fashion and written in standard English?

Reviewer #1: Yes

Reviewer #2: Yes

5. Review Comments to the Author

**Reviewer #1:**The manuscript provides a robust and well-structured exploration of host-like sequences within cryptic phage-related open reading frames in clinical isolates of Mycobacterium tuberculosis. The authors successfully integrate large-scale genomic screening, advanced protein embedding approaches (ProtBERT), and structural prediction methods (ColabFold/AlphaFold2) supported by molecular dynamics simulations. The rationale is sound, the methodology generally rigorous, and the results are clearly presented. The work contributes novel insights into phage–host genomic interplay in M. tuberculosis, a topic of growing biological and clinical interest.

Most analyses are appropriately designed and the conclusions are consistent with the data. The manuscript already meets the essential criteria for technical soundness expected by PLOS ONE. A few minor refinements would further strengthen its clarity and reproducibility. Specifically, it would be useful to include explicit accession numbers or repository links for the datasets analyzed, as well as a concise statement describing the contamination control procedures (e.g., kraken2 or CheckM screening). Providing this information will reinforce transparency and allow readers to replicate the analysis pipeline more easily.

From a writing standpoint, the paper is generally clear and fluent, though a light editorial review could improve phrasing consistency and precision in a few sections (particularly in the Discussion). The figures are well conceived, and the interpretation of structural and functional findings remains appropriately cautious.

Overall, this is a high-quality and original contribution that integrates bioinformatics, genomics, and structural biology to address a complex and underexplored aspect of M. tuberculosis evolution. With the minor clarifications above, I consider the manuscript suitable for publication in PLOS ONE.

**Reviewer #2:**This is an interesting paper that explores the potential role of phage-driven horizontal DNA transfer in Mycobacterium tuberculosis.

I have some comments and questions. As I am a bacteriologist I apologize if I miss some of the details related to the (for me complex) bioinformatics analysis. There are no line numbers which makes this process more complex.

In the title the authors include the phrase “phage-driven horizontal transfer or convergent evolution?” which ends with a question mark. In the Abstract the authors state “This divergence is driven by intragenomic recombination, phage-mediated genetic exchange, and the acquisition of diverse mutations.”. As this paper is trying to highlight the role of phages in this process and the acquisition of diverse mutations is as far as I am aware considered the primary mechanism of diversity generation in the M. tuberculosis complex I would request this phrase is adapted. For example “This divergence is driven by the acquisition of diverse mutations ,intragenomic recombination, and potentially also phage-mediated genetic exchange.”

In the introduction it is stated that “horizontal gene transfer (HGT) is rarely reported and remains a topic of debate” this is correct. Then the authors say “in contrast, cross species HGT, especially between phage and Mtb, is widely known” no citations follow this statement. Although the authors do cite literature that report the use of phages to assess the effect of antibiotics and viability etc. could the authors clarify and if available provide appropriate references after the statement “..is widely known.”

I assume the data set used is largely or possible entirely of short read Illumia data (the pre-print 35 cited mentions Illumina data). This is presumably a serious limitation with respect to accurately generating accurate de novo assemblies. I think this point warrants discussion and should be mentioned as a serious limitation.

On page 13 it is stated “Interestingly, some unknown clusters of ORFs showed similarity with distantly related species, suggesting interspecies genetic exchange.” as the data studied was derived from clinical isolates which may contain human DNA as well as DNA from other species this point is address but I think deserves more emphasis. Where these sequences reliably associated with particular lineages or strains, if so that could add support to the assumption that they are truly present in the genomes studied? This should be emphasized as a limitation.

On page t6 and elsewhere the phrase “slowed down evolution” is used. I do not agree with this statement, laboratory isolates are under pressure to adapt to laboratory media. It may be true that clinical isolates are under increased selection pressure from the host, and do not undergo transmission bottlenecks, than laboratory strains but “their evolution” is not slowed down.

Finally as some of the DNA sequences detected may represent contamination in the sequencing reads with human or (other species of)bacterial DNA would it be possible to confirm the presence of these sequences in an alternative set of clinical isolates?

6. PLOS authors have the option to publish the peer review history of their article (what does this mean?). If published, this will include your full peer review and any attached files.

Reviewer #1: No

Reviewer #2: No

---

## [Author Response · Author response to Decision Letter 1]

13 Apr 2026

Point by point response to the comments asked by reviewers

We thank the respected reviewers and editor for taking the time to review our manuscript and providing us with their valuable suggestions. We have incorporated additional information into the revised manuscript to improve it. Please find below our point-by-point response to the reviewers' comments.

Reviewer’s comments:

Reviewer #1: The manuscript provides a robust and well-structured exploration of host-like sequences within cryptic phage-related open reading frames in clinical isolates of Mycobacterium tuberculosis. The authors successfully integrate large-scale genomic screening, advanced protein embedding approaches (ProtBERT), and structural prediction methods (ColabFold/AlphaFold2) supported by molecular dynamics simulations. The rationale is sound, the methodology generally rigorous, and the results are clearly presented. The work contributes novel insights into phage–host genomic interplay in M. tuberculosis, a topic of growing biological and clinical interest.

Response: We thank the reviewer for taking the time to carefully review our manuscript and for providing valuable suggestions. Please find below our point-by-point response to the reviewer’s queries.

Reviewer #1: Most analyses are appropriately designed, and the conclusions are consistent with the data. The manuscript already meets the essential criteria for technical soundness expected by PLOS ONE. A few minor refinements would further strengthen its clarity and reproducibility. Specifically, it would be useful to include explicit accession numbers or repository links for the datasets analysed, as well as a concise statement describing the contamination control procedures (e.g., kraken2 or CheckM screening). Providing this information will reinforce transparency and allow readers to replicate the analysis pipeline more easily.

Response: We understand the reviewer’s concern about including accession numbers and repository links for the analysed dataset to ensure the reproducibility of our work. The Sequence Read Archive (SRA) accessions for the datasets used in the current study are listed in Supplementary Table 2. The genome accessions in cluster 19 are listed in Supplementary Table 3 (S3 Table), along with checkM2 results. In brief, a 0.3% (median) contamination and 100% (median) genome completeness were observed in the cluster 19 datasets.

The modified text is included in the revised manuscript:

Line 360: Further, we also performed checkM2 analysis for analyzing genome completion and contamination metrics of the cluster 19 genomes (n=971). This showed 0.3% contamination (median) with values ranging from 0.1 to 4.3% and genome completion of 100% median with values ranging from 94 % to 100 %. Individual sample details with SRA accession numbers is provided in S3 Table.

Reviewer #1: From a writing standpoint, the paper is generally clear and fluent, though a light editorial review could improve phrasing consistency and precision in a few sections (particularly in the Discussion). The figures are well conceived, and the interpretation of structural and functional findings remains appropriately cautious.

Overall, this is a high-quality and original contribution that integrates bioinformatics, genomics, and structural biology to address a complex and underexplored aspect of M. tuberculosis evolution. With the minor clarifications above, I consider the manuscript suitable for publication in PLOS ONE.

Response: We have proofread the manuscript and made appropriate changes in the revised version to ensure it sounds scientific. We thank Reviewer 1 for their careful evaluation of our manuscript, positive comments, and for highlighting the importance of our work.

Reviewer #2: This is an interesting paper that explores the potential role of phage-driven horizontal DNA transfer in Mycobacterium tuberculosis. I have some comments and questions. As I am a bacteriologist I apologize if I miss some of the details related to the (for me complex) bioinformatics analysis. There are no line numbers which makes this process more complex.

In the title the authors include the phrase “phage-driven horizontal transfer or convergent evolution?” which ends with a question mark. In the Abstract the authors state “This divergence is driven by intragenomic recombination, phage-mediated genetic exchange, and the acquisition of diverse mutations.”. As this paper is trying to highlight the role of phages in this process and the acquisition of diverse mutations is as far as I am aware considered the primary mechanism of diversity generation in the M. tuberculosis complex I would request this phrase is adapted. For example “This divergence is driven by the acquisition of diverse mutations,intragenomic recombination, and potentially also phage-mediated genetic exchange.”

Response: Thank you for providing your comment to bring clarity to the manuscript title. We have modified the title and the abstract as suggested.

In the revised manuscript, modified text is as follows:

Title: Large-scale cryptic proteome mining revealed potential phage-mediated host-pathogen genetic exchange in Mycobacterium tuberculosis.

Line 25: This divergence is driven by the acquisition of diverse mutations, intragenomic recombination, and potentially phage-mediated genetic exchange.

Reviewer #2: In the introduction it is stated that “horizontal gene transfer (HGT) is rarely reported and remains a topic of debate” this is correct. Then the authors say “in contrast, cross species HGT, especially between phage and Mtb, is widely known” no citations follow this statement. Although the authors do cite literature that report the use of phages to assess the effect of antibiotics and viability etc. could the authors clarify and if available provide appropriate references after the statement “..is widely known.”

Response: Thank you for pointing this out. We have revised the introduction and added more references to support our argument. We hope the revised text is clear.

Line 84: “Like other bacteria, evolution in Mtb is caused by several mechanisms, such as single-nucleotide polymorphisms (SNPs), insertions/deletions (INDELs), intragenomic recombination, and mycobacteriophage attacks (Chiner-Oms et al., 2019; Davies-Bolorunduro et al., 2024; Stucki & Gagneux, 2013). However, unlike many other bacteria, intra-species horizontal gene transfer (HGT) is rarely reported and remains a topic of debate due to the organism’s complex, lipid rich and relatively impermeable cell wall structure (Boritsch et al., 2016; Chiner-Oms et al., 2019). In contrast, gene families such as insertion sequences, CRISPR-associated elements, and toxin-antitoxin loci, many of which have viral origin, indicate ongoing acquisition into the Mtb genome, highlighting that its evolution is driven not only by mutations, albeit less frequentlyy, by phage-mediated gene transfer events (Shabbir et al., 2016; Villarreal, 2011; Yamada et al., 2019). This sequence transfer from phages to Mtb is leveraged to produce genetically modified Mtb strains and has been explored as a novel diagnostic approach (Fu et al., 2015; Hatfull, n.d.; Mayer et al., 2016; Piuri et al., 2009; Samaddar et al., 2015).“

Reviewer #2: I assume the data set used is largely or possibly entirely of short-read Illumina data (the pre-print 35 cited mentions Illumina data). This is presumably a serious limitation with respect to accurately generating accurate de novo assemblies. I think this point warrants discussion and should be mentioned as a serious limitation.

Response: We acknowledge that using short-read sequencing data for de novo assembly is a limitation of this study and have explicitly mentioned it in the manuscript.

Long-read sequencing data of Mtb in public repositories are primarily from direct, uncultured, or minimally processed samples. This can introduce contamination in sequencing data. Availability of Mtb short-read sequencing data is very high, and most studies used pure Mtb DNA samples from clinically isolated strains. Also, our analytical framework is sensitive to sample size, making the use of widely available cleaner short-read data more practical for robust inference.

Reviewer #2: On page 13, it is stated, “Interestingly, some unknown clusters of ORFs showed similarity with distantly related species, suggesting interspecies genetic exchange.” As the data studied were derived from clinical isolates, which may contain human DNA as well as DNA from other species, this point is addressed, but I think it deserves more emphasis. Where these sequences are reliably associated with particular lineages or strains, if so, that could add support to the assumption that they are truly present in the genomes studied? This should be emphasised as a limitation.

Response: Thank you for this valuable critique. We have now added checkM2 analysis, a reliable tool for assessing genome completeness and contamination metrics. We found a negligible 0.3% median contamination and 100% genome completeness in the genomes of cluster 19. Figure 3 displays the abundance of cluster 19 medoids in specific Mtb lineages with varying drug resistance profiles. While a detailed association analysis between these proteins and specific lineages may be conducted in future studies, we believe it is beyond the scope of the present work. We have included it as limitation of the study.

Reviewer #2: On page 6 and elsewhere, the phrase “slowed down evolution” is used. I do not agree with this statement; laboratory isolates are under pressure to adapt to laboratory media. It may be true that clinical isolates are under increased selection pressure from the host, and do not undergo transmission bottlenecks more than laboratory strains, but “their evolution” is not slowed down.

Response: Thank you for this input. Mtb laboratory strains are stored in controlled conditions, usually at -80oC since their isolation. We agree that they are also evolving, at least to the laboratory conditions. We have replaced the term “slowed down evolution” with “reduced selection pressure” in the revised manuscript. We hope that this change is satisfactory.

Line 76: However, laboratory strains of Mtb, such as H37Rv exhibit comparatively reduced evolutionary diversity due to controlled laboratory conditions and repeated passaging from the same stock.

Line 366: The present study was initially aimed to explore emerging ORFs in clinical Mtb isolates, focusing on genomic features that may be masked in laboratory strains of Mtb due to their reduced evolutionary dynamics.

Line 430: Since laboratory strain H37Rv exhibits reduced evolutionary diversity compared to clinical isolates, the findings of the present study, that is, the presence of phage domains linked with host DNA in clinical Mtb genomes, may represent evolutionary events that occurred during natural adaptation.

Reviewer #2: Finally as some of the DNA sequences detected may represent contamination in the sequencing reads with human or (other species of) bacterial DNA would it be possible to confirm the presence of these sequences in an alternative set of clinical isolates?

Response: We have run the checkM2 analysis and included the results in Supplementary Table 3. The results indicate a median of 0.3% negligible contamination across 971 cluster-19 genomes. The genomes also showed 100% genome completion.

Identical sequences may be present in different datasets, but fishing out such genomes with our sequences of interest may not be practical currently because 1) due to allopatric evolution, some different strains might have similar but not identical sequences, and 2) it would require a large-scale analysis of millions of samples. We would again like to emphasise that, due to allopatric evolution, it would be more sensible to fish out genomes with sequences of similar origin, that is, viral sequence-associated fusion ORFs. In another study currently in progress, we found evidence of fusion transcripts between Mtb genes and virus-associated genes, including insertion sequences. Kindly refer (BioRxiv, 2025, In-silico evidence of non-operonic fusion transcripts in Mycobacterium tuberculosis: Algorithm optimisation and signatures of genome plasticity, https://doi.org/10.1101/2025.09.22.677293).

The modified text is included in the revised manuscript:

Line 360: Further, we also performed checkM2 analysis for analyzing genome completion and contamination metrics of the cluster 19 genomes (n=971). This showed 0.3% contamination (median) with values ranging from 0.1 to 4.3% and genome completion of 100% median with values ranging from 94 % to 100 %. Individual sample details with SRA accession numbers is provided in S3 Table.

---

## [Editor Report · Decision Letter 1]

19 Apr 2026

Large-scale cryptic proteome mining revealed potential phage-mediated host-pathogen genetic exchange in Mycobacterium tuberculosis.

PONE-D-25-49411R1

Dear Dr. Bhalla,

We’re pleased to inform you that your manuscript has been judged scientifically suitable for publication and will be formally accepted for publication once it meets all outstanding technical requirements.

Kind regards,

Swati Jaiswal

Academic Editor

PLOS One
---

## [Editor Report · Acceptance letter]

PONE-D-25-49411R1

PLOS One

Dear Dr. Bhalla,

I'm pleased to inform you that your manuscript has been deemed suitable for publication in PLOS One. Congratulations! Your manuscript is now being handed over to our production team.

Kind regards,

on behalf of

Dr. Swati Jaiswal

Academic Editor

PLOS One